# Structural and Interactional Analysis of the Flavonoid Pathway Proteins: Chalcone Synthase, Chalcone Isomerase and Chalcone Isomerase-like Protein

**DOI:** 10.3390/ijms25115651

**Published:** 2024-05-22

**Authors:** Jacob A. Lewis, Eric P. Jacobo, Nathan Palmer, Wilfred Vermerris, Scott E. Sattler, James A Brozik, Gautam Sarath, ChulHee Kang

**Affiliations:** 1Department of Chemistry, Washington State University, Pullman, WA 99164, USA; jacob.lewis2@wsu.edu (J.A.L.); eric.jacobo@wsu.edu (E.P.J.); brozik@wsu.edu (J.A.B.); 2Department of Agriculture—Agricultural Research Service, Wheat, Sorghum, and Forage Research Unit, Lincoln, NE 68583, USA; nathan.palmer@usda.gov (N.P.); scott.sattler@usda.gov (S.E.S.); gautam.sarath@usda.gov (G.S.); 3Department of Microbiology & Cell Science and UF Genetics Institute, University of Florida, Gainesville, FL 32610, USA; wev@ufl.edu

**Keywords:** anthocyanidins, anthocyanin, flavones, flavonols, *Panicum virgatum*, *Sorghum bicolor*, metabolon

## Abstract

Chalcone synthase (CHS) and chalcone isomerase (CHI) catalyze the first two committed steps of the flavonoid pathway that plays a pivotal role in the growth and reproduction of land plants, including UV protection, pigmentation, symbiotic nitrogen fixation, and pathogen resistance. Based on the obtained X-ray crystal structures of CHS, CHI, and chalcone isomerase-like protein (CHIL) from the same monocotyledon, *Panicum virgatum*, along with the results of the steady-state kinetics, spectroscopic/thermodynamic analyses, intermolecular interactions, and their effect on each catalytic step are proposed. In addition, PvCHI’s unique activity for both naringenin chalcone and isoliquiritigenin was analyzed, and the observed hierarchical activity for those type-I and -II substrates was explained with the intrinsic characteristics of the enzyme and two substrates. The structure of PvCHS complexed with naringenin supports uncompetitive inhibition. PvCHS displays intrinsic catalytic promiscuity, evident from the formation of *p*-coumaroyltriacetic acid lactone (CTAL) in addition to naringenin chalcone. In the presence of PvCHIL, conversion of *p*-coumaroyl-CoA to naringenin through PvCHS and PvCHI displayed ~400-fold increased *V_max_* with reduced formation of CTAL by 70%. Supporting this model, molecular docking, ITC (Isothermal Titration Calorimetry), and FRET (Fluorescence Resonance Energy Transfer) indicated that both PvCHI and PvCHIL interact with PvCHS in a non-competitive manner, indicating the plausible allosteric effect of naringenin on CHS. Significantly, the presence of naringenin increased the affinity between PvCHS and PvCHIL, whereas naringenin chalcone decreased the affinity, indicating a plausible feedback mechanism to minimize spontaneous incorrect stereoisomers. These are the first findings from a three-body system from the same species, indicating the importance of the macromolecular assembly of CHS-CHI-CHIL in determining the amount and type of flavonoids produced in plant cells.

## 1. Introduction

Among the specialized plant metabolites, flavonoids are ubiquitous and have essential functions for nodulation, pathogen resistance, fertility, protection against UV radiation reactive oxygen species (ROS), attracting pollinators and microbial symbionts, and coloration [1,2,3].

Over 7000 flavonoids have been identified and classified as chalcones, flavones, flavonols, isoflavones, anthocyanidins, and anthocyanins [4]. Together with phenylpropanoids, flavonoid production may have evolved in response to stress events by serving a pivotal role during the transition from aquatic to terrestrial life [5,6]. After the core synthetic pathways for flavonoids and monolignols were established in early land plants, branches of the pathway have likely evolved to cope with changing habitats, co-evolving pathogens, and herbivores [6].

In addition, human health can benefit from flavonoids due to their antioxidant or cytotoxic properties. Although it is difficult to quantify health benefits in populations, in vivo studies have shown flavonoids can be beneficial in treating cancer, obesity, cardiovascular disease, and diabetes and can reduce the effects of aging [7,8,9,10,11]. Consequentially, there is an interest in engineering flavonoid biosynthesis, but this is challenging due to the many enzymes involved.

Chalcone synthase (CHS, EC 2.3.1.74) catalyzes the first step in generating the basic flavonoid skeleton and represents the branch point from the general phenylpropanoid pathway towards flavonoid biosynthesis (Figure 1A). CHS belongs to the family of type-III polyketide synthases (PKS) and is one of the most conserved enzymes among all land plant species [12]. Although type-I and type-II PKS form large protein complexes, type-III PKS exist uniquely as homodimers across plants, fungi, and bacteria [13]. CHS generally catalyzes multiple condensation reactions with one p-coumaroyl-CoA molecule and three malonyl-CoA molecules [14]. The reaction initiates with loading p-coumaroyl-CoA, a product of 4-coumarate-CoA lyase (4CL; EC 6.2.1.12) in the general phenylpropanoid pathway, to the catalytic cysteine in its active site. Then, three successive additions of malonyl-CoA molecules produced from primary metabolism [15] generate one tetraketide intermediate in the active site pocket of CHS. Claisen condensation reactions result in the formation of the new ring (A-ring), yielding naringenin chalcone (2′,4,4′,6′-tetrahydroxychalcone) as a product [16] (Figure 1B). It has also been shown that CHS may accept other phenylpropanoid-CoA molecules such as feruloyl-CoA and caffeoyl-CoA [17].

Naringenin and several other flavonoid molecules are known inhibitors of CHS. Specifically, naringenin acts as an uncompetitive inhibitor [18]. The accumulation of flavonoids, such as naringenin, may be toxic to the cells. Therefore, it has been proposed as a built-in feedback system in the flavonoid synthetic pathway [19].

CHS is commonly described as having “catalytic promiscuity”, giving rise to other polyketide molecules [20] such as *p*-coumaroyltriacetic acid lactone (CTAL) and bis-noryangonin (BNY). The promiscuity is not restricted to CHS because the flavonoid reductases have also been shown to accept and produce various molecules [21]. Recent studies have shown that this promiscuity of CHS can be rectified with chalcone isomerase like-protein (CHIL), which belongs to the chalcone isomerase-fold protein family [22]. Enzymatic assays conducted with CHS in the presence of CHIL reduced the formation of other polyketides, including CTAL, without altering the total amount of produced polyketide [22,23]. Therefore, CHIL has been hypothesized as an auxiliary subunit biasing CHS to generate naringenin chalcone [22]. Based on differential scanning fluorimetry, CHIL has been proposed as a metabolic sensor stabilizing thermal shifts in the presence of naringenin chalcone and destabilizing in the presence of naringenin. However, Wolf-Saxon et al. (2023) noted that there is still uncertainty concerning whether a destabilizing effect would prompt a positive or negative feedback loop.

*Chalcone isomerase* (CHI, EC 5.5.1.6) catalyzes a stereospecific intramolecular cyclization reaction for the bicyclic compound, naringenin chalcone producing tricyclic (C_6_(A)-C_3_(C)-C_6_(B)) *S*-flavanone over the *R*-isomer. In vivo, a spontaneous Michael addition reaction in solution could produce the (*2S*)-flavanone naringenin. However, in the presence of CHI, the turnover rate of naringenin chalcone to naringenin is increased by a factor of 10^7^, approaching nearly the diffusion limit [24].

CHI enzymes can be divided into four classes [25]. Type I CHI isomerizes naringenin chalcone (2′,4′,6′,4-tetrahydroxychalcone) to (*2S*)-naringenin (5,7,4′-trihydroxyflavanone) and is ubiquitous in land plants except legumes. Type II exists solely in legumes and has an activity for not only the type I reaction but also converts isoliquiritigenin (2′,4′,4-trihydroxychalcone) to the corresponding (*2S*)-5-deoxyflavonone, liquiritigenin (7,4′-dihydroxyflavanone) [26,27]. Type III comprises fatty acid-binding proteins that have a very similar three-dimensional fold as observed in type-I CHI [28]. Type IV is the aforementioned CHIL that works to rectify the promiscuity of CHS.

The formation of multienzyme complexes among the participating enzymes in the flavonoid pathway was first proposed in *Arabidopsis thaliana,* and evidence for their existence has been obtained through a yeast two-hybrid system [29]. This formation of metabolons in the flavonoid pathway may be species-specific depending on their specific needs [30]. In addition to *Arabidopsis thaliana* and *Zea mays*, the corresponding interactions have been confirmed in common hop (*Humulus lupulus*), which promotes the formation of at least a three-enzyme complex anchored by the membrane protein, prenyltransferase (PT) with CHS and CHIL [31]. In *Oryza sativa*, it was shown that CHS interacted with another cytochrome P450 enzyme, F3′H, ANS, DFR, and F3H in a yeast two-hybrid system [32]. Several studies into the complexation of the flavonoid metabolon demonstrated a weak and competitive association around an ER-anchored cytochrome P450 [33,34,35]. The formation of a metabolon through a cytochrome P450 enzyme has been observed even when the cytochrome P450 is not directly utilized in catalysis. In snapdragon, associations were observed between enzymes involved in consecutive steps, FNSII with CHS and CHI, and between nonconsecutive steps in enzymatic reactions such as FNSII and DFR [35].

Flavonoid profiles in plants are of key importance for plant fitness, but they also have applications to human health, agricultural production, and biomass refining [36,37,38]. Therefore, modifying flavonoid biosynthesis in crop plants offers prospects of greater adaptation to biotic and abiotic stresses and a benefit to consumers through reported and proposed health benefits [39]. Flavonoid levels in several plants have been altered using transgenic approaches. For example, overexpression of CHI in Chinese licorice (*Glycyrrhiza uralensis*) increased flavonoid production [40].

One of the greatest challenges in the enhanced production of flavonoids via enzymatic engineering is the lack of fundamental understanding of the structure and function of the participating enzymes in the target species [41,42]. This study comprehensively characterized CHS, CHIL, and CHI in the flavonoid pathway of two closely related monocots, sorghum and switchgrass, both of which are gaining prominence due to their resilience to weather extremes anticipated as a result of climate change. Through enzyme activity assay, molecular docking based on three crystal structures, ITC and FRET, with proteins from the same monocot species for the first time, we confirmed that both CHI and CHIL physically interact with CHS in a non-competitive manner. The primary focus was switchgrass enzyme since the flavonoid biosynthesis is induced in switchgrass in response to both abiotic and biotic stress, where they may serve protective roles within cells [43,44,45], but sorghum CHI (Bmr30) was included for comparison [46]. Since both species have a shared evolutionary history among the grasses [47], the results from switchgrass presented here can be applied to other grasses, notably other species within the Panicoideae (sorghum, maize, sugarcane, and miscanthus), but likely also to the Pooideae (wheat, barley) and Erhartoideae (rice).

## 2. Results

### 2.1. Overall Structure of PvCHS

Recombinant PvCHS (Pavir.8KG261900.1.p) expressed in *E. coli* BL21(DE3) with an N-terminal His6-tag was purified and crystallized in a ternary complex with both naringenin and CoA molecules through co-crystallization by screening the crystals with buffer containing those compounds. The corresponding diffraction data were collected at a temperature of 100 K with up to 2.04 Å resolution (Table 1, PDBID: 8V8M) from Advanced Light Source (ALS, beam line 5.0.1). The space group of the resulting complex crystal was P1. A molecular replacement approach was conducted using the coordinates of *Arabidopsis thaliana* CHS (AtCHS, PDBID: 6DXB) [6], successfully resolving residues from 8 to 395 among 402 residues in the native PvCHS. The overall structure of each subunit of PvCHS contained fifteen α-helixes and thirteen β-strands with the αβαβα core motif that is often conserved among enzymes containing thiolase-fold (Figure 2A) [13]. The crystal lattice packing indicated that four molecules in the triclinic unit cell were composed of two closely associated homodimers. The root-mean-square-deviation (RMSD) between the two homodimers was 0.181 Å, and they were assembled with a non-crystallographic two-fold symmetry. The two PvCHS subunits in each dimer established a dimer interface with 32 hydrogen bonds and 17 salt bridges, suggesting that PvCHS likely exists as a homodimer in a cellular environment. PDBePISA [48] analysis supported a high likelihood of a homodimer with a complexation significance score (CSS) of 1.0, a maximal value.

The electron density corresponding to an extended CoA molecule (Figure 2B) along a continuous tunnel piercing deeply into the PvCHS molecule was visible from an early refinement stage in only the A and C subunits. At the equivalent regions of the dimeric partners, the B and D subunits, the same tunnels were visible with only crystallographic waters near the active site. The overall structure of apo-form PvCHS was not altered significantly upon binding CoA molecules. The only noticeable difference between apo-form (B and D subunits) and CoA binary complex (A and C subunits) PvCHS was the conformation of a short loop, ^271^FHLLKD^276^ near the adenosyl and thiol groups of the bound CoA molecule, which appeared to stabilize the bent CoA confirmation described below.

A DALI Search [49] with the crystal structure of PvCHS was performed to find structurally similar proteins. The most similar structure was a chalcone synthase from *Oryza sativa* (PDBID: 4YJY) [50]. This was followed by other CHSs in descending order: *Freesia x hybrida* (PDBID: 4WUM) [4], *Arabidopsis thaliana* (PDBID: 6DXB) [6], *Medicago sativa* (PDBID: 1BI5) [51], *Malus domestica* (PDBID: 5UC5) [52], and Glycine max (PDBID: 7BUS) [53]. A BLASTP search [54,55] with the amino acid sequence of PvCHS yielded a list of similar proteins in more or less the same order as the DALI search, with OsPKS (91% amino acid identity) being the most similar, followed by AtCHS (81%), MdCHS (81%) and GmCHS (81%). The PKSs and CHSs with the highest sequence similarity showed a high degree of conservation of secondary structure (Figure 3A). A BLASTP search for PvCHS was also conducted utilizing the non-redundant sequence database, which indicated a high degree of sequence similarity (>80%) specifically among grass CHSs.

### 2.2. Substrate-Binding Pocket of PvCHS

In the A and C subunits, the adenine nucleotide and the pantoic acid moieties of CoA were bound at the exterior portion of the above-mentioned tunnel, and the sulfhydryl group at the tip of CoA molecule was located at the vicinity of the catalytic Cys-170. Notably, the extended CoA-binding tunnel is connected to another large cavity that could accommodate the substrate (malonyl CoA)/intermediates/product molecules (Figure 4A). This cavity in A and C subunits has an estimated volume of 204 Å^3^ and was connected to two surface-connected smaller branches oriented in a T-shaped manner (Figure 4A). One of the two branches was filled with crystallographic water molecules that establish a hydrogen bond network potentially linking the Cys-170 to the surface. The entrance of one port was established by Tyr-39, Tyr-42, Met-73, Met-90, Thr-197, Phe-201, Arg-202, His-208, Thr-267 and His-269. The entrance for the other branch was slightly larger and was established mainly by non-polar residues: Gly-259, Phe-268, Pro-378, Thr-135, Phe-218, Ile-257, Gly-308, and Leu-266. Notably, the apo-form subunits (B and D) did not have those two cavities.

The bound pyrophosphate group of CoA was surrounded by α12, α13, α14, and β10. The surface charge diagram displayed a positive charge from the sidechains of Lys-65, Arg-61, and Lys-58, which interacted with the pyrophosphate groups of CoA. In addition, the carbon chain proximal to the pyrophosphate of CoA was stabilized via a hydrophobic effect from the sidechains of Val-213, Leu-270, Leu-217, and Val-274. The ribosyl group of the CoA molecule adopted a C2′-endo conformation, and its pyrophosphate group established a hydrogen bond interaction with the guanidium side chain of Arg-61. An internal hydrogen bond stabilized the specific torsional angle of the CoA to N6A of the adenosine ring to a crystallographic water that was hydrogen bonded to the C5P carbonyl group that is proximal to the thiol group. The same water was also hydrogen bonded to the backbone of both Leu 270 and Leu 271. The side of PvCHS that is proximal to the expected phenolic moiety of *p*-coumaroyl-CoA was established by β5, β9, β10, and β12.

A similar location and conformation of the catalytic triad of PvCHS, Cys-170, His-309, and Asn-342, were observed as in other CHS structures. Especially, as observed in CHSs of euphyllophytes [6], the sidechain of Cys-170 displayed extra electron density and was fit with cysteine sulfinic acid, which indicates its enhanced nucleophilicity and susceptibility to oxidation. In four PvCHS subunits, the Nε2 atom of catalytic His-309 was located with a mean distance of 3.2 Å from the hydroxyl group of the modeled sulfinic acid, likely lowering p*Ka* of the catalytic Cys-170 by forming a stable imidazolium–thiolate ion pair as reported before (Appendix A) [57]. The sidechain of Asn-342 was 3.2 Å away from the hydroxyl group of sulfinic acid. In addition, Nε2 of His-309 was 3.7 Å away from the side chain Asn-342. Therefore, the sidechains of His-309 and Asn-342 could establish an oxyanion hole that stabilizes the tetrahedral transition state of the reactant.

### 2.3. Naringenin-Binding of PvCHS

To investigate the potential inhibitory role of flavonoid molecules [18,58], naringenin was added to the crystallization buffer. In addition to a bound CoA molecule, one naringenin molecule was visible in the concave pocket established at the interface between two PvCHS dimers in a stacked orientation of the A and C rings from each naringenin molecule. The B-ring of the naringenin molecules was in a nearly perpendicular orientation to their A and C rings. These naringenin molecules bound at the interface were under a vast array of hydrogen bonds linked to the nearby residues and water molecules. In detail, the hydroxyl group of B-ring was hydrogen-bonded to a water molecule that was hydrogen-bonded to the backbone of Cys-63, Asp-64, and Ile-68, and two other crystallographic waters. In addition, the C2 hydroxyl group of the C-ring was hydrogen-bonded to a water molecule that was 2.95 Å away from the carboxyl sidechain of Glu-336. This water also interacted with another crystallographic water coordinated to a hydroxyl group on the A-ring. The hydroxyl group attached to C7 on the A-ring was hydrogen bonded to a water molecule that was, in turn, hydrogen bonded to the sidechains of His-332, Arg-328, and Arg-355 (Figure 2C).

The C170S mutant PvCHS was generated to investigate its potential impact on the local conformation and confirm the catalytic role. It crystallized in an orthorhombic space group (Table 1, PDBID: 8V8N). The asymmetric unit contained two mutant PvCHS molecules, as observed in the wild-type PvCHS. Molecular replacement was conducted using the PvCHS complex structure. The overall RMSD between the mutant C170S and the WT PvCHS was small, with 0.26 Å. Importantly, the catalytic triad remained stable with a distance of 4.49 Å (3.2 Å in wild type) between the Ser-170 hydroxyl group and Asn-342 carboxamide and 5.57 Å (vs. 3.2 Å) between the serine hydroxyl and the Asn-342 carboxamide. Nε2 of His-309 was 3.84 Å (vs. 3.2 Å). The noticeable changes in distances are associated with the shorter sidechain group of serine than the oxidized cysteine.

### 2.4. Overall Structure of PvCHI and SbCHI

The apo-form of PvCHI (Figure 5A) was crystallized in a monoclinic space group, C2 (Table 1, PDBID: 8V8L). The structure was determined through a molecular replacement method using the coordinates of *Medicago sativa* CHI (AtCHI, PDBID: 1EYQ) [59]. The asymmetric unit contained two molecules of PvCHI that did not display any significant interaction between them. As predicted, PDBePISA calculated that interaction between two PvCHI molecules in the asymmetric unit was low with a Δ^i^G of 1.1 kcal mol^−1^ and a CSS of 0.00, indicating its likely monomeric nature in solution. Among 237 residues in the native PvCHI, the electron density corresponding to the residues from 4 to 217 and 4 to 216 of the A and B molecules, respectively, were resolved, indicating the disordered nature of both N- and C- terminals. The three-dimensional structure of PvCHI displayed an open-faced β-sandwich fold with six anti-parallel β-sheets (β1–β6) and seven α-helices (α1–α7).

The closely related CHI from sorghum (SbCHI, Bmr30) [46] was also crystallized in a monoclinic space group (Table 1, PDBID: 8V8P). The structure of SbCHI was determined through molecular replacement using CHI from *Deschampsia antarctica* (DaCHI, PDBID: 5YX3). The overall structure of SbCHI was very similar to that of PvCHI, with an RMSD of 0.66 Å. The noticeable differences between SbCHI and PvCHI were observed in the loops between β3a and β3b and β2 and β3a. In addition, α8 of PvCHI was shorter, as the C-terminus of PvCHI was more disordered than that of SbCHI.

To find structurally similar proteins in the PDB, a DALI search was performed utilizing PvCHI as a search model. Results indicated that CHIs from *Medicago truncatula* (PDBID: 6MS8), *Deschampsia antarctica* (PDBID: 5YX4), and *Arabidopsis thaliana* (PDBID: 4DOI) are the closest with Z-scores of 36.2, 35.7 and 35.0, respectively. Following those, several CHIs from other species including *Medicago sativa* (PDBID: 1EYQ), ancestrally inferred engineered CHIs (PDBID: 5WL4), *Saccharomyces cerevisiae* (PDBID: 6JCN), and the type III and type IV CHI from *A. thaliana* (PDBID: 4DOK, 4DOL) displayed somewhat lower Z-scores ranging from 19–30. Following these, there was a large drop-off in the Z-score to 7 for a type-II secretion chaperone (PDBID: 6O38) (Figure 3B). On the other hand, a BLASTP search using PvCHI indicated a high-level amino acid sequence identity (>80%) with other bona fide type-1 CHIs from switchgrass, *Panicum millaceum*, *Panicum hallii*, *Setaria italica, Setaria viridis*, *Digitaria exillis*, *Deschampsia antarctica*, *Oryza sativa*, and other grasses (Appendix A).

### 2.5. The Substrate-Binding Pocket of SbCHI

Our attempt to produce a binary complex with the substrate was only successful for SbCHI. From the early stage of refinement, the electron density corresponding to liquiritigenin that had been added to the crystallization buffer was clearly identified. The bound liquiritigenin, the product of isoliquiritigenin, a type II substrate, was surrounded by ^34^RGI^36^ from β3a, ^44^FTAI^47^ from β3a, ^104^YAGKVTEN^111^ from α4, and ^147^SI^148^ from α6. Notably, the molecule was surrounded with predominantly hydrophobic residues: Ile-38, Ile-36, Phe-45, and Leu-99. In addition, a hydrogen bond was established between the sidechain of Lys-107 and the C2′ hydroxyl group of the bound liquiritigenin (Appendix A).

To compare the position of bound liquiritigenin with that of the type-I substrate, naringenin chalcone was docked to the structure of PvCHI. The A-ring of the docked naringenin chalcone is at the same location as that of isoliquiritigenin in our binary complex structure of PvCHI (Figure 5C). However, their B-rings were in slightly different conformations. The B-ring of docked naringenin was superimposable with only the naringenin B-ring in the binary complex structure of MsCHI (1EYQ).

*Overall structure of PvCHIL*: The purified recombinant PvCHIL was crystallized in a tetragonal space group (Table 1, Figure 5A), and the diffraction data up to 2.98 Å were collected from ALS beamline 5.0.1 at a temperature of 100 K (Table 1, PDBID: 8V8O). Molecular replacement was performed using the coordinates of *Arabidopsis thaliana* CHIL (AtCHIL, PDBID: 4DOK) [28]. The electron densities corresponding to the residues from 6 to 21 and from 26 to 219 among each of the 223 residues in the native PvCHIL molecule were resolved. The overall secondary structure was composed of eight β-sheets and seven α-helices.

The structure of PvCHIL was very similar to that of PvCHI in terms of both the number of secondary structural elements and the overall three-dimensional structure with a backbone RMSD of 1.21 Å. The area of the largest structural differences between PvCHI and PvCHIL was located at the residues corresponding to the ligand-binding pockets of CHI. Sequence alignment between PvCHI and PvCHIL showed that amino acids constituting most of the secondary structural components are well conserved (Figure 5A). The secondary structural motifs that established the substrate-binding cleft in PvCHI were composed of α6–α7, α4–α5, β3a–β3f. These structural motifs in PvCHIL existed with only slight differences in the loops between β3C and α4 and α5 and β3D. However, the corresponding primary sequence for those secondary structural features is not conserved, as the active site is replaced in PvCHIL with the following residues: Arg-35 (Thr-37), Thr-47 (Asn-49), Asn-113 (Ser-115), Ser-190 (Trp-193) with Tyr-106 (Tyr-108) as the only conserved active site residue (Figure 5D).

A DALI search [49] with the three-dimensional structure of PvCHIL to search for structurally similar proteins yielded AtCHIL as the most similar structure, with a Z-score of 31.9, followed by the engineered ancestral CHI (PDBID: 5WL8) with Z-scores ranging between 27 and 28. Following these were CHI from *Medicago sativa* (6MS8) with a Z-score of 26.3, *Arabidopsis thaliana* (4DOI) with a Z-score of 25.6, *Deschampia antarctica* (5YX3) with a Z-score of 25.6. This was followed by the type III CHI from *Arabidopsis thaliana* (4DOL), with a significant drop in the Z-score congruent with the unique fold observed in CHI. Despite lower sequence-level identity, less than 60% was observed in *bona fide* type-1 CHI; a BLASTP search with PvCHIL identified several type IV monocot CHIs. Ancestrally inferred CHI displayed roughly 50% sequence identity. Bona fide type 1 CHIs were identified with sequence identities <30% (Appendix A).

### 2.6. Activity of CHS with and without CHIL

The enzyme kinetics of PvCHS were probed with *p*-coumaroyl-CoA as the starter molecule and excess malonyl CoA as the extender molecule, as described before [22]. The same experiments were conducted in the presence and absence of PvCHIL (Figure 6A, Table 2). The observed data were analyzed utilizing a Michaelis–Menten model and a Hill plot. For the assay conducted in the presence of PvCHIL, the better fit was to the Hill plot with a higher R2 value. The *V_max_* for *p*-coumaroyl-CoA was 0.0505 μM s^−1^ and 0.00695 μM s^−1^ in the presence and absence of PvCHIL, respectively, indicating a 7.7-fold increase of *V_max_* in the presence of PvCHIL (Figure 6B, Table 2). The Khalf of the Hill equation was 1.315 μM in the presence of PvCHIL, and the slope of the Hill plot was 0.67, indicating negative cooperativity. In the absence of PvCHIL, the *K_m_* of the Michaelis–Menten equation was 0.130 µM.

### 2.7. Activity of CHI

To investigate the substrate preference of PvCHI, an enzymatic assay was performed based upon previous assays with minor changes [60], allowing cyclization of either naringenin chalcone or isoliquiritigenin, which are the type I and type II substrates, respectively. The reaction rates for varying concentrations of naringenin chalcone displayed a typical Michaelis–Menten kinetics with a *V_max_* of 8012 μM s^−1^ and a *K_m_* of 16.04 μM (Figure 7A, Table 2). Significantly, the activity of PvCHI for isoliquiritigenin displayed a substrate inhibition pattern with a *V_max_* of 0.6344 μM s^−1^, a *K_m_* of 17.60 μM, and a *K_I_* of 11.20 µM (Figure 7A, Table 2). For comparison, the steady-state kinetics of SbCHI were also investigated with the same substrates. The *V_max_* of SbCHI was 2889 μM s^−1^ and 0.238 μM s^−1^ for naringenin chalcone and isoliquiritigenin, respectively. The *K_m_* was 16.55 μM and 1.585 μM for naringenin chalcone and isoliquiritigenin, respectively.

To investigate a potential inhibition effect, steady-state kinetics of type-I substrate naringenin chalcone were also investigated in the presence of the type-II product liquiritigenin in PvCHI. Although the presence of 2.5 µM and 5 µM liquiritigenin did not affect the *V_max_*, the *K_m_* was increased in a liquiritigenin concentration-dependent manner, indicating a competitivity between the type I and type II substrates (Figure 8). The *K_I_* of PvCHI for isoliquiritigenin was 0.285 µM.

### 2.8. Activity of CHS with CHI and CHIL

A combined activity assay with both CHS and CHI in the reaction mixture, which monitored the conversion of *p*-coumaroyl-CoA to naringenin and naringenin chalcone, was conducted. Specifically, kinetic parameters were compared for the reaction Mixture I containing only 5× molar excess of PvCHI to PvCHS, and the reaction Mixture II containing 5× molar excess of both PvCHI and PvCHIL to PvCHS. Reaction Mixture I produced naringenin with a *V_max_* of 0.175 μM s^−1^ (Figure 6, Table 2). Although analysis with LC/MS-MS did not detect any naringenin chalcone, CTAL was observed in the reaction mixture (Appendix A). Reaction Mixture II did not produce any quantifiable amount of naringenin chalcone, but the CTAL peak was largely diminished. Instead, the production of naringenin in Mixture II was observed with a ~400-fold increased *V_max_* value of 3.040 μM s^−1^. The values for the *K_m_* of PvCHS for *p*-coumaroyl-CoA were not affected in the presence of PvCHI and PvCHIL.

### 2.9. Analysis of Interactions among CHS, CHI, and CHIL

Several enzymes in the flavonoid pathway are reported to form a complex [33]. However, some of those interactions have been reported to be highly species-specific [61]. Specifically, CHIL interacts with CHS, potentially serving as an activator of these enzymes in *Arabidopsis thaliana* and some other plant species. To investigate these kinds of interactions among CHS, CHI, and CHIL from identical species, several experiments were performed using switchgrass enzymes.

### 2.10. Isothermal Titration Calorimetry

To probe the interaction between PvCHIL and PvCHS, isothermal titration calorimetry (ITC) was applied as performed to the closely related enzymes in the same pathway [21]. The interaction between PvCHS and PvCHIL was endothermic with ΔH of 31.88 kcal mol^−1^, ΔS of 127 cal mol^−1^ deg^−1^, and *K_d_* of 76.3 µM (Table 3, Appendix A). To further probe the possibility of PvCHIL as a biological sensor in this interaction, both naringenin and naringenin chalcone, which are the substrate and product of CHI, respectively, were added in 5× molar excess equivalents compared to the concentration of CHS and CHIL. In the presence of naringenin and naringenin chalcone, the *K_d_* was 7.46 µM and 170.35 µM, respectively, indicating that in the presence of naringenin, affinity between PvCHIL and PvCHS is increased, whereas affinity between them is decreased in the presence of naringenin chalcone.

### 2.11. Fluorescence Resonance Energy Transfer (FRET)

Although the interaction between CHS and CHIL has been reported in several plant species [22], CHI has only been shown to interact with CHS in *snapdragon* (*Antirrhinum majus*) and *A. thaliana* [29,35]. Previous research has identified CHIL as a rectifier for the promiscuous enzymatic activity of CHS [30,31,35,62,63]. Thus, a potential interaction among three proteins from an identical plant, switchgrass, was investigated with the appropriate FRET pairs. In all cases, ATTO 488 was used as the donor dye. ATTO 550 and ATTO 647N were both used as acceptor dyes, where the ATTO 488/ATTO 550 pair has a shorter Förster radius than the ATTO 488/ATTO 647N pair. The *K_d_* for PvCHS to PvCHI was 637 ± 357 nM (Figure 9A,B), while the *K_d_* for PvCHS to PvCHIL was 26.53 ± 21.0 µM (Figure 9C,D and Appendix A). The FRET spectra for PvCHI and PvCHS displayed an isosbestic point at 580 nm, thus indicating a simple one-stage interaction between PvCHI and PvCHS. On the other hand, PvCHIL and PvCHS lacked a clear isosbestic point, which means the interactions between PvCHI and PvCHS could have an intermediate state. A three-body system of CHS, CHI, and CHIL was also tested to determine if PvCHI and PvCHIL displayed a competitive nature in binding to PvCHS. The three-protein system utilized the same dye attached to PvCHS previously, ATTO 488. PvCHI was labeled with ATTO 647N, and unlabeled PvCHIL was held constant. The *K_d_* of PvCHI was unaffected by the addition of PvCHIL, indicating that the binding site is non-competitive (Appendix A).

### 2.12. Substrate Molecular Docking

Substrate molecular docking of the apo-form PvCHI crystal structure with the type I substrate of CHI, naringenin chalcone, and the type II substrate of CHI, isoliquiritigenin, were performed. The docked positions of both substrates were located at the previously identified cleft but slid along the α4 helix away from the active site and close to the enzyme’s surface. Upon superimposition of PvCHI with MsCHI (PDBID: 1EYQ), the position of the entire α4, which constitutes one side of the substrate-binding pocket and displayed high-temperature factors, were significantly different, and the cleft in the PvCHI was narrower than in MsCHI (Appendix A). Molecular docking was tried again after repositioning of α4 of PvCHI to the equivalent position of MsCHI by adjusting the ψ and φ angles of the N-terminal Gly-103 of α4. After adjusting the position of α4, the docked position of naringenin chalcone was superimposable with the observed location of naringenin in MsCHI (PDBID: 1EYQ). The closed conformation of α4, which narrows the binding cleft, is consistently observed in all deposited CHIs from dicots but not DaCHI and PvCHI.

Geometry optimization for both naringenin chalcone and isoliquiritigenin was conducted using ORCA 5.0.4 to identify the lowest energy conformation of both the protonated and unprotonated forms of both molecules by allowing adjustment of all dihedral angels, bond angles, and bond distances [64]. The molecular energy was monitored by rotating the dihedral angle around the bond between the γ-carbonyl carbon, C1′, and the α, β unsaturated carbon bond (Figure 10). The calculation was completed using a def2-QZVPP basis set [65]. The protonated naringenin chalcone in its lowest energy displayed a proper conformation for cyclization reaction with its 2′ hydroxyl group distanced 2.75 Å from the β carbon of the target double bond at a dihedral angle of 14.4°. However, the minimum of the deprotonated form was 4.27 Å between reactive atoms with a dihedral angle of 69.8° in an improper conformation for a Michaelis addition reaction, which indicates the protonated form is likely preferentially bound to the active site of CHI. In contrast, the distance from the 2′ hydroxyl group to the β carbon was 5.35 Å and 3.31 Å for the protonated and deprotonated forms of isoliquiritigenin, respectively, which requires further adjustment for Michaelis addition reaction for both forms.

### 2.13. Molecular Docking between PvCHIL and CHS, PvCHI, and PvCHS

Protein–protein docking was conducted using the HDOCK Server [66] to postulate their plausible interaction. A score below −200 indicates a high likelihood of an intermolecular interface in solution. PvCHS showed a high likelihood of interaction with PvCHIL and PvCHI, as the corresponding docking scores were −240 and −210, respectively. The PvCHS:PvCHIL interface was established by 39 residues from PvCHS and 27 residues from PvCHIL. Close inspection of the results indicated that the PvCHIL utilized its unique protruding loop between β3a and β3b to bind near the homodimer interface of PvCHS (Figure 11). Nine hydrogen bonds were identified between the A subunit of PvCHS and PvCHIL, and only single hydrogen bond between the B subunit PvCHS and PvCHIL. Additionally, the sidechains of Arg-185 and Arg-234 on the A subunit of the PvCHS dimer established a salt-bridge with the sidechain of Asp-37, Glu-39, and Asp-79 of PvCHIL. In addition, the sidechains of Asp-179 and Arg-284 of the B subunit of PvCHS interact with those of His-41 and Glu-101 of PvCHIL.

On the other hand, the PvCHI molecule was docked on the homodimer interface of CHS but on the opposite side of the PvCHIL-binding site. The PvCHS-PvCHI interaction involved 27 residues from PvCHS and 23 residues from PvCHI. The interactions at the interface were predominantly hydrophobic, with only two salt bridges involving Arg-262 and Glu-263 from the B-subunit of PvCHS and Glu-38 and Lys-109 from PvCHI. Notably, the docked site of PvCHI was proximal to the exit tunnel of the PvCHS active site and was roughly 15 Å from the exit to the general PvCHI active site. Significantly, a tighter interaction for both PvCHI: PvCHS and PvCHIL: PvCHS was noticed in subunits A and C than in subunits B and D, which suggests a possible allosteric effect of *p*-coumaroyl CoA.

## 3. Discussion

The potential interaction and formation of complexes among enzymes in the flavonoid synthetic pathway has been proposed in several species. The impact of the supramolecular complex on flavonoid formation through substrate-channeling is theorized to be dynamic and in response to environmental shifts [67]. Thus, a comprehensive understanding of each participating enzyme of the flavonoid synthetic pathway and the complex formation among them can lead to changes in the flux through the pathway to favor specific compounds.

### 3.1. Catalytic and Inhibition Mechanisms of PvCHS

In our crystal structures of PvCHS, a double oxidative state was observed at the catalytic Cys-170 residue, as previously identified in euphyllophyte CHS (Appendix A). The sulfinyl group in Cys-170 probably resulted in a more nucleophilic residue with a lower *pK_a_* could enhance resistance toward competing oxidation reactions by H_2_O_2_ and other reactive oxygen species in the cell [6]. Another catalytic residue, His-309, was sandwiched between this Cys-170 and Cys-344 with its imidazole Nε2 and Nδ1 atoms at a distance of 3.2 Å and 4.1 Å from the sidechain sulfur of Cys-170 and Cys-344, respectively (Appendix A). The presence of Cys-344 could help maintain the imidazoline–thiolate ion pair, also lowering the pKa of Cys-170 [6]. The observed large cavity that is connected to the extended CoA tunnel not only captures the p-coumaroyl moiety by hydrogen bond interaction with Gly-222 and Asp-223 but also holds the *p*-hydroxyl group at a pivot point during three consecutive decarboxylative condensation reaction of malonyl CoA. During these elongation steps with repeated loading of malonyl CoA, His-309 and Asn-342 stabilize their transition state by forming an oxyanion hole supported by previous kinetic data [68]. In these processes, the noticeable asymmetric distribution of hydrophobic residues in the pocket established by Ile-257, Phe-218, Phe-268, and Pro-378 could guide and stabilize a proper conformation of the growing ketide chain.

The presence of only one CoA molecule per homodimer and the disappearance of two small channels in the apo-form subunit of PvCHS indicate potential cooperativity between two subunits. To support this hypothesis, only a single CoA molecule was also observed previously in the homodimer of *Huperzia serrata* PKS1 that shares a high similarity with PvCHS (PDBID: 3AWJ). Likewise, the cooperativity was observed in the steady state kinetics of PvCHS being fitted with a Hill plot (Figure 6A), which is consistent with that of AtCHS [22]. Therefore, it is possible that the two subunits of CHS in some species may act in tandem with only single subunit functions per turnover.

Naringenin has long been considered an uncompetitive inhibitor of CHS [18]. However, the naringenin binary complex structures of MsCHS (1CGK) and GmCHS (7BUR) supported competitive inhibition due to the location of the associated naringenin near or at the active site. Notably, the position of naringenin in our binary complex structure of PvCHS was away from the active site differing from previous studies [51,53], providing a structural inference that flavonoids could act as uncompetitive inhibitors (Figure 2C). These naringenin molecules located in the CoA complexed PvCHS could correspond to a proper uncompetitive inhibition mode as reported before for various flavonoids [18,19]. This type of interaction could establish a feedback inhibition system, given that the accumulation of flavonoids is known to be toxic to plant cells [19].

### 3.2. Binding and Catalytic Mechanism of PvCHI

When the crystal structures of MsCHI complexed with naringenin (PDBID: 1EYP) are superimposed with our PvCHI structure, the α4 helix of apo-form PvCHI, which constitutes a major portion of the substrate-binding pocket, was shifted away from the α4 position of MsCHI (Appendix A). In addition, in the apo-form crystal structures of both PvCHI and SbCHI, the sidechain of Arg-35 in PvCHI (Arg-34 in SbCHI) was positioned at the substrate-entrance site and hydrogen bonded to both Thr-47 and Tyr-190 through water-molecules. Therefore, to dock naringenin into the same pocket of PvCHI, both adjustments of the α4 helix and Arg-34 were required. Noticeably, in the structure of MsCHI, the naringenin binary complex (PDBID:1EYQ), the sidechain of the Arg-35 is rotated to the opposite side and interacts with Asp-200 located in α7. The residue corresponding to aspartate in DaCHI, PvCHI, MtCHI, and AtCHI is alanine, eliminating the possibility of salt-bridge formation in those CHIs. Differing from the previously reported mechanism [59], the R34M or R34A mutant DaCHI (Arg-35 in PvCHI) displays the most reduced activity among extensive mutations for all the residues establishing a substrate-binding pocket [60], suggesting its critical role in the activity of CHI. The observed temperature factors for residues 35–39 and those constituting the α4 helix in PvCHI were elevated compared to other residues, indicating the intrinsic flexibility that allows the substrate to enter with a plausible induced fit. High-temperature factors were observed around Arg-35 in PvCHI and were also observed in both MsCHI (PDBID: 1EYQ) and DaCHI (PDBID: 5YX3). In addition to its gating role for substrate entrance in combination with flexible α4 residues, Arg-35 likely participates in catalysis by lowering the *pK_a_* of Thr-47. Alternatively, upon association of substrate, the water molecules in the substrate-binding pocket are pushed out, creating a low dielectric environment at the active sites where electrostatic interactions become much stronger. Consequently, the positively charged guanidium sidechain of Arg-35 induces deprotonation of the bound naringenin chalcone. This ionization of the 2′-hydroxyl group into the 2′-oxyanion (*pK_a_* ~7–8) enables the attack of the α, β-unsaturated double bond of naringenin chalcone utilizing a Michael addition. Our conformational analysis of protonated naringenin chalcone (Figure 10) shows a preference for the 2′-hydroxyl group to be in the proper position of 2.79 Å from the new bond formed through the Michael addition. Still, the most stable conformation of the deprotonated one is not proper for the same reaction. After readjusting the α4 helix and Arg-34, the minimum potential conformer of protonated naringenin chalcone could be docked to the same site of PvCHI, but the most stable deprotonated form of naringenin chalcone could not. Thus, it is likely that the enzyme preferentially binds the protonated form of naringenin chalcone in its lowest energy state. In this specific conformation, its 2′-hydroxyl group is in proper orientation for a Michael addition reaction.

### 3.3. Type II Activity of PvCHI

The substrate specificity of type I and II CHIs has previously been attributed to the difference between Ser-190 and Ile-191 in type II vs. Thr and Met, respectively, in type I [59,60]. PvCHI has Ser-190/Ile-191 in the α6 region and displayed significant activity toward isoliquiritigenin (Appendix A). Type I and type II substrates have been identified at slightly different locations along the binding cleft. In DaCHI, the binding position of isoliquiritigenin (PDBID: 5YX4) matches closely with that observed in this study. A thorough investigation of the neighboring amino acid residues of both the DaCHI, PvCHI, and SbCHI structures showed that the observed orientation and position of isoliquiritigenin are likely not conducive to a cyclization reaction. Our kinetic data indicated that liquiritigenin acts as a competitive inhibitor of naringenin chalcone in PvCHI. Therefore, the binding location for type-I and type-II substrates might be similar in the enzymes containing both type-I and II activities, such as DaCHI, PvCHI, and SbCHI. The *pK_a_* of the lone *ortho*-hydroxyl group of isoliquiritigenin was calculated with MolGPKa to be 7.3, 0.6 units higher than the *pK_a_* of 6.7 for naringenin chalcone [69]. Not only is the *pK_a_* of the hydroxyl group higher, but geometry optimization of isoliquiritigenin also indicates that the distance between the same hydroxyl group and the α,β-unsaturated double bond is 4.49 Å for the protonated form versus 4.80 Å for the deprotonated form. The combination of those two factors may be the reason for the decreased activity of PvCHI for isoliquiritigenin. Considering that CHI’s specificity and near-diffusion limit activity is based on capturing a specific populated substrate with a proper conformation, engineering an altered specificity might be challenging.

The *K_m_* values of PvCHI for the type-I and type-II substrates were very similar, indicating that observed in vivo competition could be due to the given cellular concentration of isoliquiritigenin or naringenin chalcone. However, the *V_max_* for isoliquiritigenin was 0.24 μM s^−1^ compared to that of naringenin chalcone, 3.1 mM s^−1^. These findings are congruent with the previous ones, establishing that type-II CHI evolved early, following the colonization of land, and not later as in the legume family [70].

### 3.4. Affinity and Effect of CHIL/CHI on the CHS

In this report, the interaction between PvCHS and PvCHI/PvCHIL and its effect on the activity of PvCHS was investigated. As previously observed in snapdragon (*Antirrhinum majus*) CHS [22], promiscuity of PvCHS was rectified by PvCHIL, yielding much less CTAL and BNY. Moreover, adding PvCHIL to the PvCHS reaction mixture also enhanced the production of naringenin chalcone with 1.7-fold increased *V_max_* but did not affect the *K_m_* of *p*-coumaroyl-CoA.

Recent findings in *Vitis vinifera* identified a modulation effect of CHIL on CHS in the presence of either flavonoids or chalcones, suggesting that CHIL could act as a biosensor via thermostability effects [23]. Our ITC results indicated that naringenin increased the affinity between PvCHS and PvCHIL ~10-fold, while naringenin chalcone decreased the affinity by ~two-fold (Table 3). As a regulatory system, the increased levels of naringenin serve as positive feedback by tightening the affinity of CHIL for CHS and thus ensure the generation of naringenin chalcone from CHS.

The PvCHS kinetic assay in the presence of both PvCHIL and PvCHI drastically increased the *V_max_* of *p*-coumaroyl-CoA to naringenin by over 400-fold with the *V_max_* increasing from 0.417 μM min^−1^ to 182.4 μM min^−1^. To probe the location of protein–protein interactions, PvCHIL and PvCHI crystal structures were docked separately from that of PvCHS [66]. The results suggested that the unique protruding loop between the β3a and β3b of CHI and CHIL interacts with the opposite side of the dimeric interface of PvCHS (Figure 9). Because PvCHS and PvCHIL interact with PvCHS on the opposite side, there is no apparent competition between CHI and CHIL also indicated by our FRET results. In addition, considering the binding site of PvCHI was oriented towards a plausible product exit site of PvCHS, it is tempting to speculate that PvCHI/PvCHS complex formation allows for direct tunneling of naringenin chalcone from CHS to CHI for stereospecific cyclization without diffusion. This scenario could be linked to a positive feedback mechanism of naringenin, as it increases the affinity between PvCHS and PvCHIL and feeds the correct stereoisomer to CHI. Likewise, the reduced affinity between PvCHS and PvCHIL in the presence of naringenin chalcone could act as a negative feedback mechanism to prevent incorrect cyclization when cellular levels of naringenin chalcone are high. The lack of an isosbestic point in the PvCHIL–PvCHS interaction FRET spectra indicates the possibility of a multistage binding model. A larger confirmational change of CHS might be necessary to ensure catalytic promiscuity is resolved upon binding of PvCHIL, which supports the hypothesis of CHIL acting as a rectifier of CHS.

The catalytic efficiency and equilibrium constant of PvCHI must be near the diffusion rate and in the forward direction, as evidenced by the absence of naringenin chalcone in both PvCHS/PvCHI and PvCHS/PvCHI/PvCHIL systems. However, other stereoisomers, such as CTAL and BNY, were detected only in the PvCHS/PvCHI system, probably due to promiscuous CHS activity in the absence of PvCHIL. The results of the enzymatic assays support a non-competitive interaction between PvCHI and PvCHIL, as the velocity was increased in the three-protein system compared with the two-protein system. Observation of CTAL in two-protein studies has been observed, even though CTAL is not observed in vivo. Thus, the combination of the interaction of both CHIL and CHI may be necessary to fully rectify the promiscuity of CHS. These findings indicate the importance of the macromolecular assembly of CHS-CHI-CHIL in determining the amounts and types of flavonoids and related pigments produced in plant cells.

## 4. Materials and Methods

### 4.1. Chemicals and Software

Analytical-grade chemicals were obtained from Sigma-Aldrich (St. Louis, MO, USA), Thermo Fisher (Waltham, MA, USA), and Alfa-Aesar (Ward Hill, MA, USA). Screening solutions for crystallization were obtained from Hampton Research (Aliso Viejo, CA, USA). Molecular graphics images were produced using the ChimeraX package (University of California San Francisco, NIH P41 RR-01081). The plotted figures were generated by GraphPad Prism V8 (GraphPad Software, Inc., Boston, MA, USA). OriginX was used for analysis of ITC data (OriginLab Corporation, Northampton, MA, USA). Structural alignments were made using Jalview (University of Dundee, Dundee, UK). *P*-Coumaroyl-CoA was obtained via enzymatic reaction using Pv4CL in a 0.1 M potassium phosphate buffer at pH 7 with 0.2 mM *p*-coumaric acid, 0.2 mM CoA, and 2.5 mM ATP. The reaction proceeded for 8 h with the continual addition of CoA, ATP, and 4CL. A RediSep C18 Gold 5 ng column was primed according to instructions, and the resulting reaction was filtered onto the column. The column was washed extensively with 4% (*w*/*v*) ammonium acetate and 2 column volumes of water. The *p*-coumaroyl-CoA was eluted using methanol. The concentration was quantified using the molar extinction coefficient at 333 nm 21,000 L mol^−1^ cm^−1^.

### 4.2. Expression and Purification of Recombinant PvCHS and Recombinant CHIs

Complementary DNA (cDNA) representing *Panicum virgatum Pavir.8KG261900* was modified to encode an N-terminal 6×-His tag and was cloned into a pET-30a (+) vector (EMD Millipore, St. Louis, MO, USA) for overexpression. The plasmid was introduced into *Escherichia coli* BL21(DE3) cells (EMD Millipore, St. Louis, MO, USA). Five mL of lysogeny broth with 50 µg mL^−1^ kanamycin was inoculated with cells from a single colony and incubated overnight prior to being transferred to 2 × 1.5 L lysogeny broth with 50 µg mL^−1^ kanamycin at 37 °C in an orbital shaker until OD_600_ = 0.6. The temperature was then reduced to 25 °C, and isoprophylthio-β-galactoside was added to a final concentration of 0.5 mM. Cells were induced for 12 h and harvested using centrifugation at 8000× *g* for 15 min at 4 °C. Buffer A (50 mM tricine, pH 7.3, 300 mM NaCl) was used to suspend cells prior to sonication on ice for 30 min (model 450 Sonifier, Branson Ultrasonics, Brookfield, CT, USA). The homogenized fraction was applied to centrifugation at 31,000× *g* for 1 h. The supernatant was immediately applied to a nickel-NTA column (Qiagen, Germantown, MD, USA) and washed with Buffer A containing 20 mM imidazole using 10-column volumes. The recombinant protein was eluted using buffer A containing 250 mM imidazole. The eluted protein was buffer exchanged to Buffer B (10 mM tricine, pH 7.3, 50 mM NaCl, and 5% (*v*/*v*) glycerol) after concentration using an Amicon 8050 ultrafiltration cell (30-kD molecular weight cutoff membrane, EMD Millipore), followed by overnight dialysis using a 30-kD membrane bag into Buffer B. The protein was applied at a flow rate of 3 mL min^−1^ to a Mono-Q Column (GE Healthcare) pre-equilibrated with Buffer B. The protein was eluted using Buffer B containing 100 mM NaCl. The resulting fraction was concentrated and injected into a HiLoad 16/600 Superdex 200 column (Cytiva) using Buffer C (10 mM Tricine pH 7.3 50 mM NaCl) at a 1 mL min^−1^ flow rate. The presence of PvCHS in eluted fractions was confirmed using SDS-PAGE, and fractions were pooled and concentrated to 10 mg mL^−1^ using Protein Assay Dye (Bio-Rad). Purity was estimated to be greater than 99%. CHS C170S mutant was generated utilizing cDNA and modified with an N-terminal 6×-His tag and was cloned into a pET-30a (+) vector (EMD Millipore, St. Louis, MO, USA) for overexpression. The purification was identical to that of the wild type.

Complementary DNA (cDNA) representing *Panicum virgatum Pavir.8KG261900* and *Pavir.8KG022400* (*PvCHIL*) and *Pavir.9KG390224* (*PvCHIL*) was modified to encode an N-terminal 6×-His tag and was cloned into a pET-30a (+) vector (EMD Millipore, St. Louis, MO, USA) for overexpression. The PvCHS plasmid was introduced into *Escherichia coli* BL21(DE3) cells (EMD Millipore, St. Louis, MO, USA), and PvCHI(L) plasmids were introduced into *Escherichia coli* Rosetta 2 cells (EMD Millipore, St Louis, MO, USA). Five mL of lysogeny broth with 50 µg mL^−1^ kanamycin was inoculated with cells from a single colony and incubated overnight prior to being transferred to 2 × 1.5 L lysogeny broth with 50 µg mL^−1^ kanamycin at 37 °C in an orbital shaker until OD_600_ = 0.6. For PvCHI and PvCHIL, chloramphenicol was also added in a 1:1000 (*v*/*v*) ratio. The temperature was then reduced to 25 °C, and isoprophylthio-β-galactoside was added to a final concentration of 0.5 mM. Cells were induced for 12 h and harvested using centrifugation at 8000× *g* for 15 min at 4 °C. Buffer A (50 mM tricine, pH 7.3, 300 mM NaCl) was used to PvCHS cultures to resuspend cells prior to sonication on ice for 30 min (model 450 Sonifier, Branson Ultrasonics). In the case of PvCHI(L), Buffer D was used to resuspend cells (50 mM Tris, pH 8, 300 mM NaCl). The homogenized fraction was applied to centrifugation at 31,000× *g* for 1 h. The supernatant was immediately applied to a nickel-NTA column (Qiagen, Germantown, MD, USA) and washed with Buffer A or D containing 20 mM imidazole using 10-column volumes. The recombinant protein was eluted using Buffer A containing 250 mM imidazole. The eluted protein was buffer exchanged to Buffer B (10 mM tricine, pH 7.3, 50 mM NaCl, and 5% (*v*/*v*) glycerol) for CHS or Buffer E (10 mM Tris, pH 8, 50 mM NaCl, and 5% (*v*/*v*) glycerol) for CHI(L) after concentration using an Amicon 8050 ultrafiltration cell (30-kD molecular weight cutoff membrane, EMD Millipore, St Louis, MO, USA), followed by overnight dialysis using a 30-kD membrane bag into Buffer B. The protein was applied at a flow rate of 3 mL min^−1^ to a Mono-Q Column (GE Healthcare, Chicago, IL, USA) pre-equilibrated with buffer B. The protein was eluted using Buffer B containing 100 mM NaCl. The resulting fraction was concentrated and injected into a HiLoad 16/600 Superdex 200 column (Cytiva) using Buffer C (10 mM Tricine pH 7.3 50 mM NaCl) for CHS or Buffer F (10 mM Tricine pH 7.3 50 mM NaCl) for PvCHI(L) at a 1 mL min^−1^ flow rate. The presence of proteinin eluted fractions was confirmed using SDS-PAGE, and fractions were pooled and concentrated to 10 mg mL^−1^ for CHS and 30 mg mL^−1^ for PvCHI(L)using Protein Assay Dye (Bio-Rad, Hercules, CA, USA). Purity was estimated to be greater than 99%. CHS C170S mutant was generated utilizing cDNA and modified with an N-terminal 6×-His tag and was cloned into a pET-30a (+) vector (EMD Millipore, St. Louis, MO) for overexpression. The purification was identical to that of the wild type.

Sobic.001G035600 (SbCHI) was modified as reported previously [46] and purified in the same manner as PvCHIL using *Escherichia coli* Rosetta 2 cells (EMD Millipore, St Louis, MO, USA).

### 4.3. Crystallization and Structure Determination

*PvCHS*: For crystallization of PvCHS, a solution of pure PvCHS (10 mg mL^−1^) in 10 mM Tricine pH 7.3 and 50 mM NaCl was prepared. Crystallization attempts were performed using the hanging-drop vapor-diffusion method at a temperature of 277 K. PvCHS crystals were obtained by mixing the above protein solution (1 μL) with an equal volume of reservoir solution containing 0.2 M sodium formate and 20% (*v*/*v*) PEG 3350 with 10 molar excesses of naringenin and CoA. Diffraction-quality crystals usually appeared after 2 days, and larger rod-shaped crystals with dimensions of approximately 2 mm × 0.5 mm × 0.5 mm were obtained after 7 days. Diffraction data were collected at Advanced Light Source in Beamline 5.0.1 at 100 K and processed with HKL2000 [71]. Molecular replacement was conducted utilizing *Medicago sativa* CHS, and refinement was conducted utilizing Phenix [72] with visualization utilizing Coot [73].

*PvCHI*: Crystallization of PvCHI was accomplished by taking 30 mg mL^−1^ enzyme in 10 mM Tris.HCl pH 8.0 and 50 mM NaCl and screening utilizing the sitting drop vapor-diffusion method at a temperature of 277 K. PvCHI crystals appeared in 10 days. The crystals were obtained with a mixture of 0.2 μL of the enzyme solution mixed with 0.2 μL of 0.2 M MgCl_2_, 0.1 M Tris: HCl pH 8.5, and 30% (*v*/*v*) PEG 4000. Data were collected at Advanced Light Source in Beamline 5.0.3 at 100 K. Data were scaled utilizing HKL2000. Molecular replacement was conducted utilizing *Deschampia antarctica* CHI, and refinement was conducted utilizing Phenix [72] with visualization utilizing Coot [73].

*PvCHIL*: Crystallization of PvCHI was accomplished by taking 10 mg mL^−1^ enzyme in 10 mM Tris.HCl pH 8.0 and 50 mM NaCl and screening utilizing the sitting drop vapor-diffusion method at a temperature of 277 K. Crystals appeared via sitting drop vapor-diffusion roughly 5 days after mixing 0.2 μL of the enzyme solution with 0.2 μL of 1.1 M ammonium tartrate dibasic pH 7.0. Data were collected at Advanced Light Source in Beamline 5.0.3 at 100 K. Data were scaled utilizing HKL2000. Molecular replacement was conducted utilizing *Arabidopsis thaliana* CHIL, and refinement was conducted utilizing Phenix [72] with visualization utilizing Coot [73].

### 4.4. Kinetic Assay of PvCHS

Kinetic assays of PvCHS followed the same methodology as previously used with slight modifications [22]. In brief, PvCHS was held at 0.1 µM with 300 µM malonyl-CoA. *p*-Coumaroyl-CoA was varied from 0–75 µM in a total reaction volume of 100 µL. In some cases, PvCHIL was included in a 5:1 molar ratio with PvCHS. The reaction proceeded for 10 min and was quenched with the addition of 10 µL of glacial acetic acid and was extracted twice using ethyl acetate and dried under a vacuum. The resulting product was dissolved in 50 µL methanol. Analysis of the resulting products was conducted utilizing a Waters Xevo TQ-MS mass spectrometer interfaced with Acquity UPLC (Waters, UK) and an Ace Excel 1.7 SuperC18 (P/N EXL-1711-1003U, 100 mm × 3.0 mm i.d.) with solvent A being water and 0.05% (*w*/*v*) formic acid and solvent B being ACN with 0.05% (*w*/*v*) formic acid at a flow rate of 0.2 mL min^−1^ over a 19 min gradient. The UV-vis spectrum (210–500 nm) of the eluted fractions was collected and analyzed with the previously mentioned TQ-MS operating in negative ionization mode utilizing multiple reaction monitoring (MRM). The capillary voltage was 2.50 kV with a cone voltage of 34 V and a temperature of 623 K for desolvation and collision gas flow of 550 L h^−1^ of helium. The collision energy was 18 V. An ion with *m*/*z* 271.000 was selected as the precursor for CTAL, naringenin chalcone, and naringenin. The selected product ion for CTAL was *m*/*z* 145.000, and *m*/*z* 151 for naringenin chalcone and naringenin. Retention times for CTAL, naringenin chalcone, and naringenin were 6.07, 9.78, and 10.31 min, respectively. MRM transitions and appropriate collision energies for the chalcone and flavonoid derivatives were found on the MassBank of North America website: https://mona.fiehnlab.ucdavis.edu/ (accessed on 12 June 2023). The resulting peaks were integrated using TargetLynx data analysis software 4.2 (Waters Corp, Milford, MA, USA). Using a linear regression, the resulting data were analyzed using GraphPad Prism 8.0.2 (La Jolla, CA, USA).

### 4.5. Kinetic Assay of PvCHI

Kinetic PvCHI assays utilized the previously reported methodology [60]. In brief, 0.1 nM of PvCHI was assayed in 50 mM potassium phosphate for its activity against naringenin chalcone and isoliquiritigenin at 390 nm using reported molar extinction coefficients [24] to track the disappearance of the substrates. The resulting data was analyzed using GraphPad Prism 8.0.

### 4.6. Isothermal Titration Calorimetry

Purified PvCHS and PvCHIL were dialyzed to PBS. Substrates were dissolved in the same buffer used for enzyme dialysis. PvCHS was placed into the well of the ITC (Malvern Worcestershire, UK), and PvCHIL was added to the syringe. A total of 22 injections were made, each with a volume of 2 µL. For injections utilizing naringenin chalcone and naringenin, they were added at a 5× molar excess to the titrant. Data were analyzed utilizing Origin Pro (Irvine, CA, USA).

### 4.7. Molecular Docking

Optimization of small molecules for docking was initially conducted utilizing ORCA [64] using a def2-SVP as the basis set for optimization and was viewed for use in Avogardo [74]. Initial screening of the enzyme was conducted by establishing a grid of 50 Å × 50 Å × 50 Å in Autodock Tools [75]. The screening was conducted using a grid centered at the center of the protein 100 as the search exhaustiveness. The resulting positions identified the binding pocket previously reported from MsCHI as the highest affinity location. Therefore, a 40 Å × 40 Å × 26 Å pocket was established at the points −13.419, 6.846, and 1.2, and the highest affinity binding position was located. Protein–protein docking coordinate files were submitted to the HDOCK server [66,76].

### 4.8. FRET

The proteins were labeled with FRET forming ATTO N-hydroxy succinimidyl (NHS)-ester dyes, ATTO 647N, and ATTO 488 [77]. The NHS-ester dyes reacted with available amino groups in the protein. The protein first underwent dialysis in Phosphate-Buffered Saline (PBS) of pH 8.4 for a minimum of 8 h at 4 °C. This deprotonated the amino groups available; doing so allowed the NHS-ester dye to react with the free amine and attach the dye to the protein. To achieve an average degree of labeling (DOL) of 2–3, the dye was added to the protein solution at threefold molar excess, followed by nutating overnight. The buffer was swapped during the protein-dye reaction three times every 6–8 h with PBS pH 8.4 and swapped one last time to PBS pH 7.4 for storage and the measurements. Once a DOL of 1 or higher was achieved, the labeled protein was ready for FRET measurements. The protein labeled with the FRET donor, ATTO 488, was held at a constant concentration of 1 uM. In contrast, the protein with the FRET acceptor, ATTO 647N, was added at concentrations starting from 0.5 uM up to 10 uM and increased incrementally by 1 uM.

The FRET measurements were performed using a home-built luminescence spectrophotometer. A 470 nm fiber-coupled LED (ThorLabs: M470F4) was used to excite the samples. The beam was passed through a 470 nm hard-coated bandpass filter (Thorlabs: FBH470-10) and then focused onto the sample using a 25 mm focal length lens. The fluorescence was collected perpendicular to the excitation beam by a collimating lens and focused on the slits of an Acton 500i Monochromator with a f-matched focusing lens (f-number = 6.5). The dispersed light was measured using a thermoelectrically cooled Hamamatsu R943-02 photomultiplier tube (PMT). The signal from the PMT was then passed through a wide-band preamplifier (SRS model SR445) and fed into a photon counter (SRS model SR400). The monochromator and photon counter were interfaced through a custom LabVIEW program. All spectra were an average of three scans and analyzed and plotted utilizing IgorPro software (8.0.4.2 (build 34722), WaveMetrics Inc., Portland, OR, USA).

## 5. Conclusions

Flavonoids are phenolic compounds that protect plants against pathogens and UV radiation and help attract pollinators. Several of these compounds have been of interest as nutraceuticals and anti-cancer drugs. Switchgrass (*Panicum virgatum* L.) and sorghum (*Sorghum bicolor* (L.) Moench) are two crops that can be cultivated with limited inputs, including on less product lands. The flavonoids in sorghum grain [78] and the abundant biomass of both switchgrass and sorghum present an opportunity to obtain tailored chemicals for medicinal and food use and/or to promote human health. The production of these value-added products also offers the prospect of increasing the value of the crop, especially because many of these compounds are soluble and readily extracted. This study characterized the enzymes involved in the first two committed steps of the flavonoid pathway, CHS and CHI, and the non-catalytic protein CHIL. Through crystal structure, molecular docking, mutagenesis, kinetic analyses, ITC, and FRET analysis, our study provides evidence supporting a revised catalytic reaction mechanism for monocot CHS and CHI and identified an intermolecular interaction of CHS with both CHI and CHIL in a non-competitive manner. In the three-protein system of CHS, CHI, and CHIL, CHIL rectified CHS activity with a drastic increase in the production of flavonoids. These findings can be utilized for the metabolic engineering of grasses with enhanced flavonoid biosynthetic efficiency through the branch point enzyme CHS and alter metabolic flux through the monolignol and flavonoid pathways to enable the production of novel bio-based products.

## Figures and Tables

**Figure 1 ijms-25-05651-f001:**
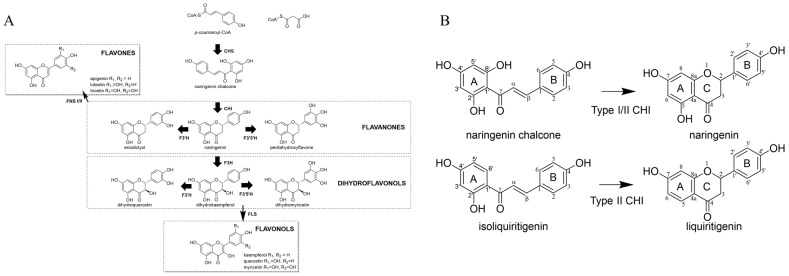
Flavonoid pathway. (**A**) General overview of the flavonoid pathway. (**B**) Type I (naringenin chalcone) and Type II (isoliquiritigenin) and their corresponding products with IUPAC numbering. Molecular graphics were created utilizing Chemdraw 22.2.0 (Waltham, MA, USA).

**Figure 2 ijms-25-05651-f002:**
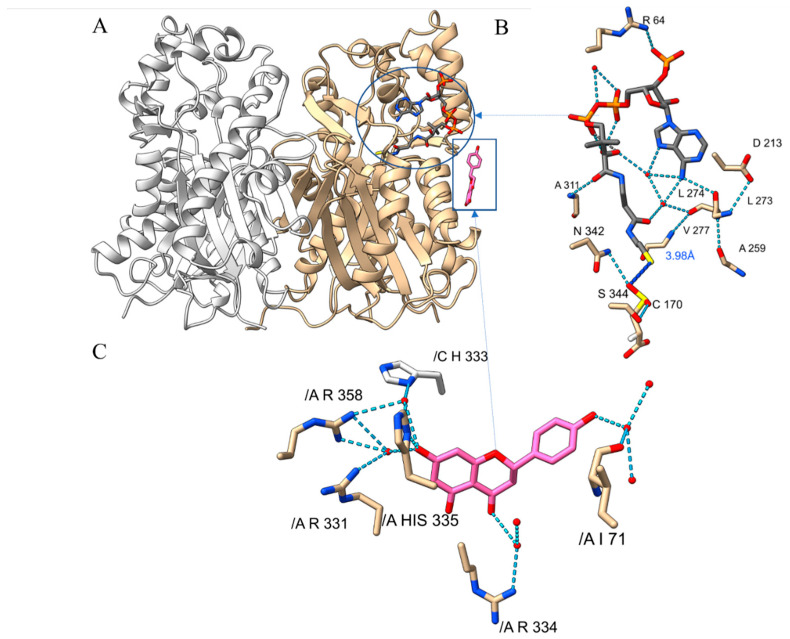
The crystal structure of PvCHS. (**A**) Ribbon diagram representing the overall structure and secondary structures of PvCHS homodimer (tan and white color for each subunit). One naringenin molecule (pink and boxed) at the surface and a CoA molecule (grey and circled) at the active site were depicted by the stick models. The locations of N- and C-terminal structural elements are labeled. (**B**) A bound CoA (gray) molecule in the active site of PvCHS with the sulfinic acid labeled as CSD, the location where the polyketide is assembled. Hydrogen bonds are shown with dotted lines and crystallographic waters are represented as red spheres. (**C**) Naringenin (pink) molecule at the surface of PvCHS. Chain identification is listed in C (as /C) to discern A chain (as /A). Molecular graphics images were produced using the ChimeraX package (UCSF).

**Figure 3 ijms-25-05651-f003:**
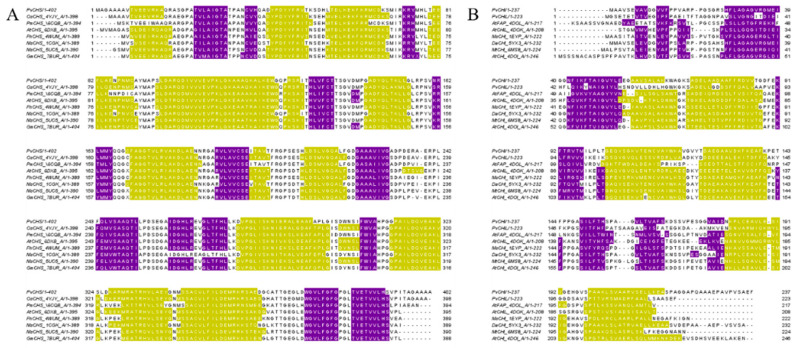
Alignment of the amino acid sequences of enzymes deposited in PDB that share a high degree of similarity. (**A**) CHS with PvCHS (PDBID: 8V8M), OsCHS (PDBID: 4YJY), PmCHS (PDBID: 6CQB), AtCHS (PDBID: 6DXB), FhCHS (PDBID: 4WUM), McCHS (PDBID: 1CGK), MsCHS (PDBID: 5UC5), and GmCHS (PDBID: 7BUR); (**B**) CHI/CHIL with PvCHI (PDBID: 8V8L), PvCHIL (PDBID: 8V8O), AtFAP (PDBID:4DOL), AtCHIL (PDBID:4DOK), MsCHI (PDBID:1EYP), DaCHI (PDBID:5YX3), MtCHI (PDBID:6MS8), AtCHI (PDBID:4DOI). The regions of α-helices are shown in purple, and β-strands are shown in yellow. The alignment was created using BLASTP to search for similar enzymes, aligned using Clustal Omega, and visualized using Jalview (University of Dundee, Dundee, UK).

**Figure 4 ijms-25-05651-f004:**
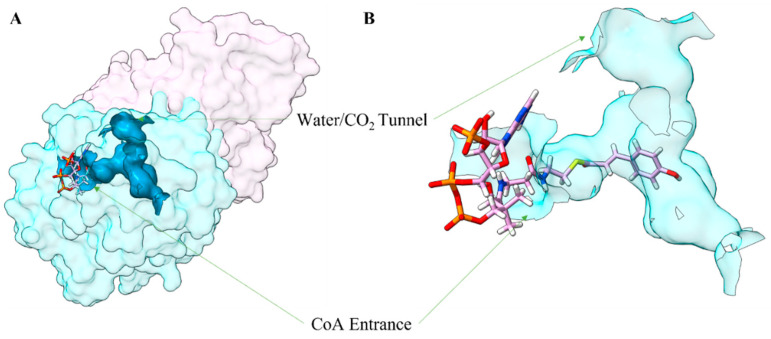
The binding Pocket of PvCHS. (**A**) Space-filling representation of a PvCHS homodimer with the substrate binding pocket shown in blue. Different subunits are shown in different colors. Calculations were made using CASTp3.0 [56], and the figure was made using ChimeraX (UCSF). (**B**) Zoomed-in view of the substrate binding tunnel with *p*-coumaroyl-CoA. Molecular graphics images were produced using the ChimeraX package (UCSF).

**Figure 5 ijms-25-05651-f005:**
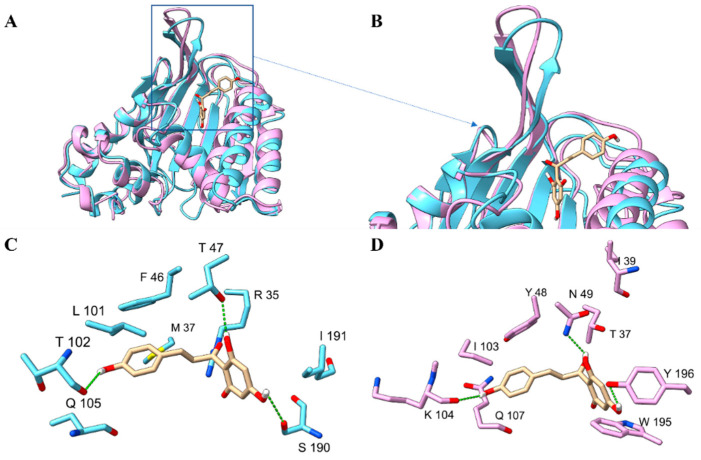
The structure of PvCHI and PvCHIL. (**A**) Ribbon diagram representing the three-dimensional structures of PvCHI (blue) and PvCHIL (pink). Two disordered loops in the crystal structure PvCHIL are shown as a dotted line. (**B**) The active site, as identified via molecular docking, is superimposed into PvCHI and PvCHIL. (**C**) Residues from PvCHI that interact with the superposition naringenin chalcone at the active site. (**D**) Residues from PvCHIL that are in the active site near the superimposed naringenin chalcone. Different active site residues: PvCHI F46 (Y48 PvCHIL), T47 (N49), R35(T37), M37 (I39), L101 (I103), T104 (K104), Q105 (Q107), S190 (W195), and I191 (Y196). Molecular graphics images were produced using the ChimeraX package (UCSF).

**Figure 6 ijms-25-05651-f006:**
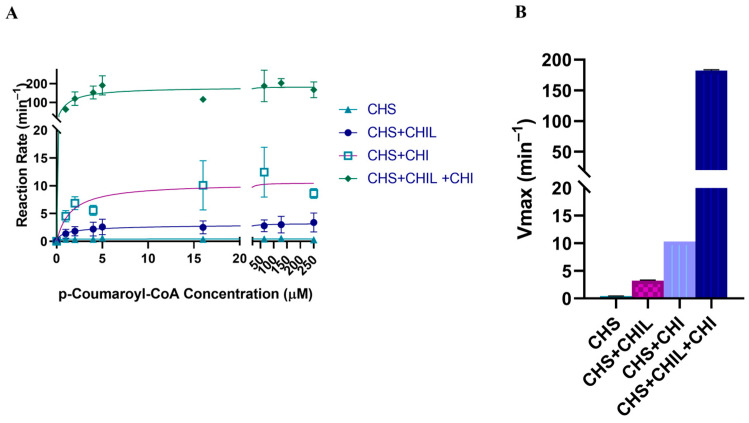
Steady-state kinetics of PvCHS with varying concentrations of *p*-coumaroyl-CoA and fixed concentration of malonyl-CoA. The product formation is treated as the sum of naringenin and naringenin chalcone. (**A**) Michaelis-Menten curve comparing PvCHS (turquoise), PvCHS and PvCHIL (dark blue), PvCHS and PvCHI (purple), and PvCHS, PvCHIL and PvCHI (green). The *K_m_* remained largely unaffected in all assays at 1 µM. The *V_max_* for PvCHS by itself was 0.417 μM min^−1^; for PvCHS with PvCHIL, it was 7.7-fold higher; for PvCHS and PvCHI, it was 24-fold higher, and for the three-protein system, it was 437-fold higher. (**B**) The *V_max_* for PvCHS (turquoise), PvCHS and PvCHI (purple), PvCHS and PvCHI (light blue), and PvCHS, PvCHIL, and PvCHI (dark blue) represented in a bar graph with the error in the calculated V_max_ by regression shown at the 95% confidence interval. Graphics and calculations were conducted utilizing GraphPad Prism 8.0.2 (San Diego, CA, USA).

**Figure 7 ijms-25-05651-f007:**
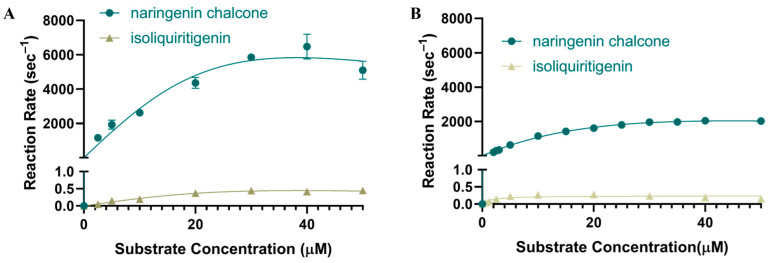
Steady-state enzyme kinetics of CHI. (**A**) PvCHI utilizing both naringenin chalcone (blue) and isoliquiritigenin (gold). The *V_max_* was 8012 µM s^−1^ and 0.63 µM s^−1^, respectively. The *K_m_* was 16.04 µM for naringenin chalcone and 17.60 µM for isoliquiritigenin. (**B**) Steady-state kinetics of SbCHI utilizing both naringenin chalcone (blue) and isoliquiritigenin (gold). The *V_max_* was 2929 µM s^−1^ and 0.24 µM s^−1^, respectively. The *K_m_* was 16.98 µM for naringenin chalcone and 1.58 µM for isoliquiritigenin. Graphics and calculations were conducted utilizing GraphPad Prism 8.0.2 (San Diego, CA, USA).

**Figure 8 ijms-25-05651-f008:**
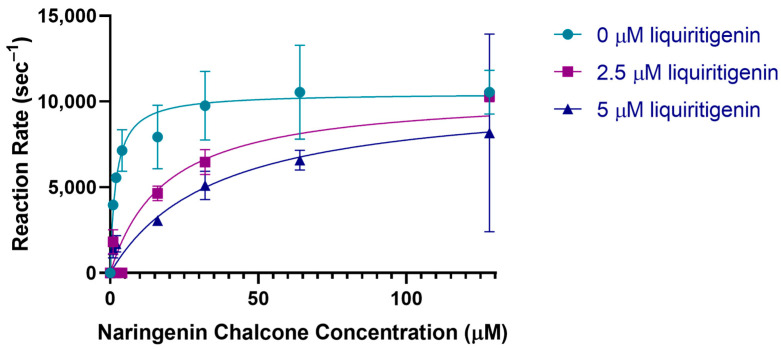
PvCHI inhibition by liquiritigenin. PvCHI enzyme kinetics for naringenin chalcone in the presence of liquiritigenin at 0 (turquoise), 2.5 µM (purple), and 5 µM (blue). Error increased at 150 µM concentration of naringenin chalcone due to saturation of the detector. Graphics and calculations were conducted utilizing GraphPad Prism 8.0.2 (San Diego, CA, USA).

**Figure 9 ijms-25-05651-f009:**
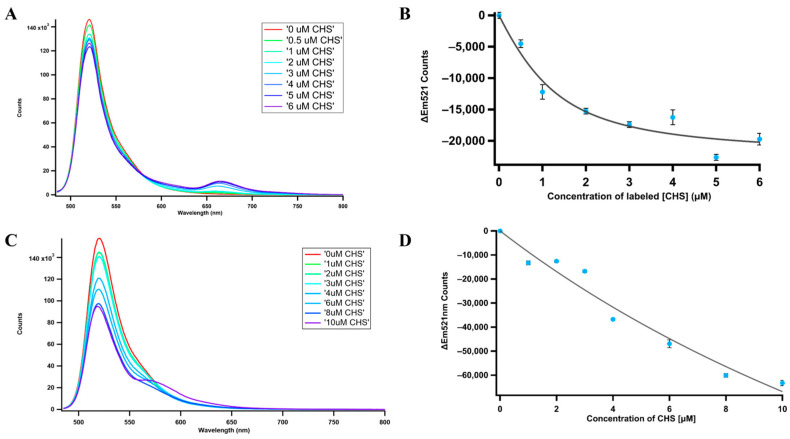
Fluorescence resonance energy transfer (FRET). (**A**) The concentration of PvCHI was held constant at 1 µM while PvCHS concentration was gradually increased. PvCHI was labeled with ATTO 488 NHS-ester dye, causing a peak at 521 nm. PvCHS was labeled with Atto 647N NHS-ester dye, causing an emission peak at 655 nm. The spectra for each concentration were taken in triplicate (*n* = 3). (**B**) The emission of PvCHI labeled with ATTO 488 NHS-ester was titrated with PvCHS labeled with ATTO 647N. The plot was then fitted with the Appendix A, and the *K_d_* resulted in 637 ± 357 nM. (**C**) The concentration of PvCHIL was held constant at 1 µM while PvCHS concentration was gradually increased. PvCHIL was labeled with ATTO 488 NHS-ester, causing a peak at 521 nm. PvCHS was labeled with ATTO 550 NHS-ester dye, causing a peak at 575 nm. The spectra for each concentration were taken in triplicate (*n* = 3). (**D**) The emission of PvCHIL labeled with Atto 488 NHS-ester was titrated with PvCHS labelled with ATTO 550 NHS-ester. The plot was then fitted with the Appendix A, and the *K_d_* resulted in 26.5 ± 21.0 µM.

**Figure 10 ijms-25-05651-f010:**
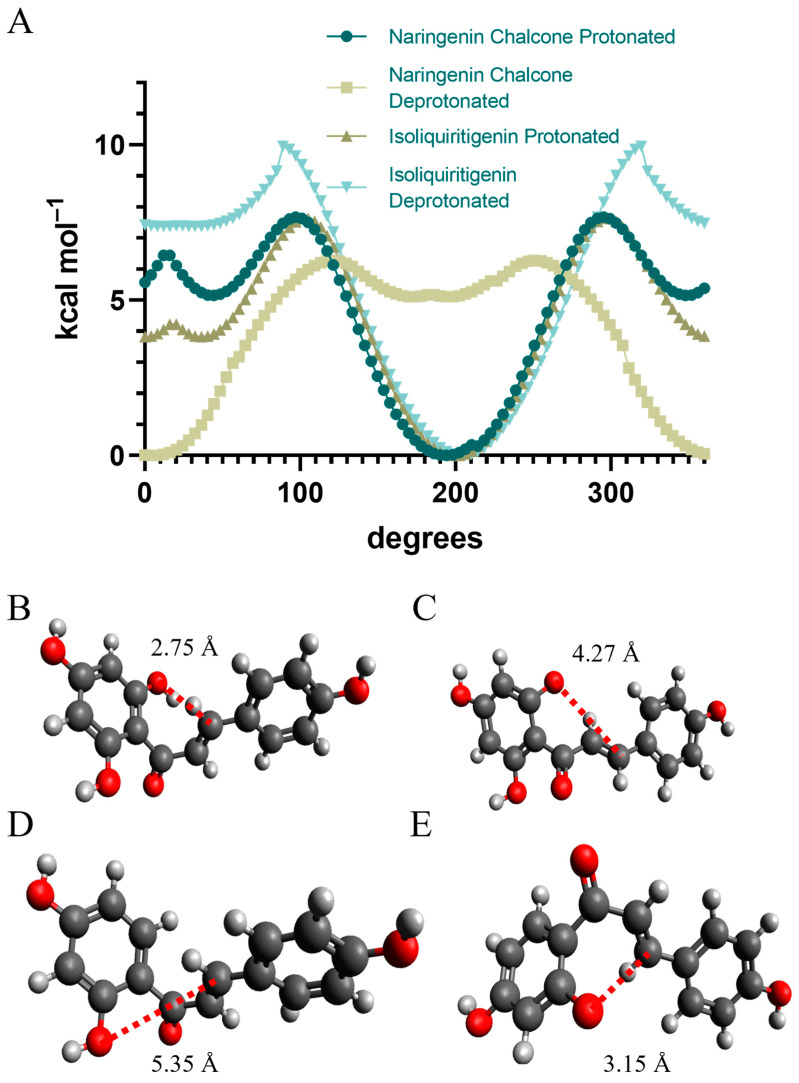
Energy scan for dihedral angle and the minimal energy conformation for naringenin and isoliquiritigenin. (**A**) The energy profile for the dihedral angle between the A and B rings of the chalcones was scanned. Optimization was performed using a def2-TZVP basis set in ORCA [64]. A distance between atoms involved in Michael addition is shown in red for (**B**) naringenin chalcone, (**C**) deprotonated naringenin chalcone, (**D**) isoliquiritigenin, (**E**) deprotonated isoliquiritigenin.

**Figure 11 ijms-25-05651-f011:**
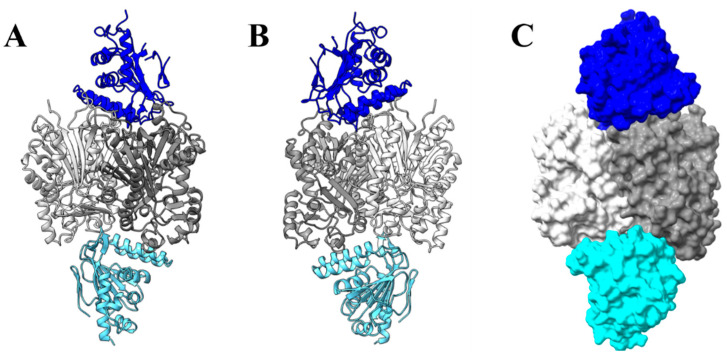
Interaction among PvCHS, PvCHI, and PvCHIL. (**A**) PvCHIL (dark blue) and PvCHI (light blue) were docked to the dimeric form of PvCHS (grey) using the HDOCK server [66]. The most favorable binding state indicates interaction at opposite sides of PvCHS. (**B**) Interaction flipped 180 degrees. (**C**) Space-filling interaction of the complex. Molecular graphics images were produced using the ChimeraX package (UCSF).

**Table 1 ijms-25-05651-t001:** X-ray diffraction data and refinement statistics for PvCHS, PvCHI, SbCHI, and PvCHIL.

	PvCHI	PvCHS (CoA and Naringenin Complex)	PvCHS C170S	PvCHIL	SbCHI (Liquiritigenin Complex)
PDBID	8V8L	8V8M	8V8N	8V8O	8V8P
Wavelength	1Å	1Å	1Å	1Å	1Å
Resolution range	36.39–1.74 (1.802–1.74)	46.59–2.29 (2.372–2.29)	39.33–2.124 (2.2–2.124)	41.92–3.21 (3.325–3.21)	46.2–3.18 (3.294–3.18)
Space group	C 1 2 1	P 1	P 21 21 21	P 43 21 2	P 1 21 1
Unit cell a, b, c (Å) α, β, γ (°)	139.09 46.521 59.054 90 106.372 90	51.05 83.085 106.45 79.431 76.253 77.211	55.757 61.722 204.087 90 90 90	100.171 100.171 118.93 90 90 90	92.453 127.825 95.311 90 116.271 90
Total reflections	120,786 (11,568)	121,592 (11,993)	148,323 (12,456)	130,626 (11,969)	112,695 (10,017)
Unique reflections	36,857 (3640)	71,129 (6990)	39,995 (3675)	10,355 (998)	33,435 (2812)
Multiplicity	3.3 (3.2)	1.7 (1.7)	3.7 (3.4)	12.6 (11.8)	3.4 (3.0)
Completeness (%)	98.27 (98.27)	95.94 (94.15)	97.62 (91.99)	98.65 (98.52)	92.07 (84.88)
Mean I/sigma (I)	11.90 (1.23)	6.64 (3.23)	6.06 (1.51)	7.19 (1.88)	5.48 (1.61)
Wilson B-factor	27.14	9.93	25.96	54.38	39.17
R-merge	0.05583 (0.7246)	0.07506 (0.1528)	0.1643 (0.6189)	0.3151 (1.119)	0.1969 (0.5945)
R-meas	0.06661 (0.8691)	0.1062 (0.2161)	0.1955 (0.7378)	0.3284 (1.171)	0.2344 (0.7241)
R-pim	0.03587 (0.4737)	0.07506 (0.1528)	0.1034 (0.3917)	0.09113 (0.3398)	0.126 (0.4087)
Wavelength (Å)	1.000	1.000	1.000	1.000	1.000
CC1/2	0.998 (0.615)	0.959 (0.954)	0.968 (0.619)	0.984 (0.68)	0.958 (0.613)
CC*	0.999 (0.873)	0.989 (0.988)	0.992 (0.874)	0.996 (0.9)	0.989 (0.872)
Reflections used in refinement	36,812 (3640)	70,877 (6981)	39,715 (3650)	10,263 (998)	30,826 (2812)
Reflections used for R-free	1991 (197)	1017 (100)	1999 (184)	1027 (100)	1880 (175)
R-work	0.1959 (0.2770)	0.2014 (0.2400)	0.2095 (0.2973)	0.2407 (0.3088)	0.2165 (0.3072)
R-free	0.2396 (0.3288)	0.1954 (0.2434)	0.2112 (0.2666)	0.2539 (0.3232)	0.2735 (0.3836)
CC (work)	0.955 (0.758)	0.932 (0.903)	0.954 (0.804)	0.921 (0.684)	0.913 (0.780)
CC (free)	0.930 (0.756)	0.936 (0.872)	0.962 (0.831)	0.944 (0.765)	0.872 (0.621)
Number of non-hydrogen atoms	3400	12,930	6320	3234	12,441
Macromolecules	3127	11,866	5932	3234	12,346
Ligands	28	136	14	0	155
Solvent	261	928	382	0	0
Protein residues	426	1554	778	418	1679
1RMS (bonds)	0.006	0.008	0.003	0.003	0.005
RMS (angles)	0.75	0.99	0.52	0.59	0.55
Ramachandran favored (%)	99.05	96.35	96.64	92.2	90.2
Ramachandran allowed (%)	0.95	3.32	2.71	7.07	8.36
Ramachandran outliers (%)	0	0.33	0.65	0.73	1.44
Rotamer outliers (%)	0.31	2.55	1.91	1.46	0.16
Clashscore	7.18	12.25	11	7.31	26.06
Average B-factor	34.94	15.41	29.02	53.66	44.46
Macromolecules	34.5	15.06	28.99	53.66	44.54
Ligands	50.72	28.16	42.95		34.12
Solvent	39.54	17.99	29.4		

r.m.s.d., Root-mean-square deviation.

**Table 2 ijms-25-05651-t002:** Kinetic parameters of PvCHI, SbCHI, and PvCHS with enzymatic partners.

Enzyme	Substrate	Km (µM)	kcat (s^−1^)	kcat Km^−1^ (s^−1^ µM^−1^)
PvCHI	Naringenin chalcone	16.04 ± 6.28	8012 ± 1185	499.5
	Isoliquiritigenin	17.60 ± 5.09	0.6344 ± 0.07195	0.03579
SbCHI	Naringenin chalcone	16.55 ± 2.08	2889 ± 143.9	174.5
	Isoliquiritigenin	1.585 ± 0.929	0.2387 ± 0.0274	0.1506
PvCHS	*p*-coumaroyl-CoA	0.1303 ± 0.0319	0.00695 ± 0.00077	0.0533
PvCHS + CHI	*p*-coumaroyl-CoA	1.545 ± 1.061	0.175 ± 0.017	0.1133
PvCHS + CHIL	*p*-coumaroyl-CoA	1.321 ± 0.263	0.0505 ± 0.0018	0.0382
PvCHS + CHI + CHIL	*p*-coumaroyl-CoA	1.121 ± 0.561	3.040 ± 0.261	2.711

**Table 3 ijms-25-05651-t003:** Thermodynamic properties of the interaction between PvCHS/SbCHIL and various ligands measured by isothermal titration calorimetry.

Enzyme	Substrate	*K_d_* (µM)	∆H (kcal mol^−1^)	∆S (cal mol^−1^ K^−1^)
PvCHS	PvCHIL	76.3	31.84 ± 7.44	127
	PvCHIL with Naringenin	7.46	5.34 ± 0.51	41.4
	PvCHIL with naringenin chalcone	170.35	36.38 ± 3.68	140

## Data Availability

The data that support the findings of this study are available from the corresponding author upon reasonable request.

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
