# Peer review of "Structural and Interactional Analysis of the Flavonoid Pathway Proteins: Chalcone Synthase, Chalcone Isomerase and Chalcone Isomerase-like Protein"

_ijms, 2024, doi:10.3390/ijms25115651_

Round 1

Reviewer 1 Report

Comments and Suggestions for Authors

In this manuscript, the authors present a comprehensive investigation into the structural and interactional aspects of flavonoid biosynthetic proteins: CHS, CHI, and CHIL. The study's findings robustly support its conclusions and demonstrate rigorous design and effective execution. Overall, the study was well-designed and effectively executed. Here are a few comments:

1. There are many repetitions in the author's article, and the content should be properly simplified. Such as ‘4.2 Expression and Purification of Recombinant PvCHS’ and ‘4.3 Expression and Purification of Recombinant CHIs’. The content is almost similar.

2. Primers for constructing vectors are not shown.

3. The clarity of some appendix figures, such as Fig S3B, is significantly compromised. Enhancing the resolution of these images.

4. ‘The primary focus was on switchgrass enzymes due to their greater amenability for crystallization’. Although this is a valid reason for selection, I believe the authors should further emphasize the significance of studying this species (Panicum virgatum) in the introduction.

5. Having ten figures in the main text may be excessive; I suggest relocating some of these figures to supplementary materials to streamline the main text.

Author Response

Reviewer 1

In this manuscript, the authors present a comprehensive investigation into the structural and interactional aspects of flavonoid biosynthetic proteins: CHS, CHI, and CHIL. The study's findings robustly support its conclusions and demonstrate rigorous design and effective execution. Overall, the study was well-designed and effectively executed. Here are a few comments:

  1. There are many repetitions in the author's article, and the content should be properly simplified. Such as ‘4.2 Expression and Purification of Recombinant PvCHS’ and ‘4.3 Expression and Purification of Recombinant CHIs’. The content is almost similar.

The PvCHS and CHI have been purified following the similar procedures.

As suggested by the reviewer, the manuscript has been revised and restructured to remove any redundancy.

  1. Primers for constructing vectors are not shown.

The corresponding section has been rewritten to clarify. The gene mutation was commercially ordered through Genscript inc.

  1. The clarity of some appendix figures, such as Fig S3B, is significantly compromised. Enhancing the resolution of these images.

In the revised manuscript, Fig. S3B has been modified to enhance the resolution.

  1. ‘The primary focus was on switchgrass enzymes due to their greater amenability for crystallization’. Although this is a valid reason for selection, I believe the authors should further emphasize the significance of studying this species (Panicum virgatum) in the introduction.

The corresponding sentence has been modified to clarify and the following three references were inserted.

  1. Having ten figures in the main text may be excessive; I suggest relocating some of these figures to supplementary materials to streamline the main text.

Several figures have already been reduced and moved to supplementary. Thus, we wish to keep the current figures in the main text.

Reviewer 2 Report

Comments and Suggestions for Authors

Please add some recent reference in L126 about how flavonoids helps crop plants to cope biotic and abiotic stresses please read and include (https://doi.org/10.3389/fpls.2023.1265700).

Please also add novelty statement in the end of introduction.

Please add recent references in method section (Isothermal titration calorimetry and FRET).

Please improve discussion by addition of some recent relevant references supporting your findings. Please also highlight your finding in discussion section and at the end of discussion add a short future perspective.

Comments on the Quality of English Language

Minor editing required

Author Response

Reviewer 2

Please add some recent reference in L126 about how flavonoids helps crop plants to cope biotic and abiotic stresses please read and include (https://doi.org/10.3389/fpls.2023.1265700).

In the revised manuscript, new references are added including the suggested one.

- https://doi.org/10.1094/PBIOMES-01-20-0013-FI

- https://doi.org/10.3389/fpls.2022.1019266

- https://doi.org/10.3389/fpls.2023.1265700).

- Dynamic Reconfiguration of Switchgrass Proteomes in Response to Rust (Puccinia novopanici) Infection

(2023) International Journal of Molecular Sciences, 24 (19), art. no. 14630, .

- Differential defense responses of upland and lowland switchgrass cultivars to a cereal aphid pest (2020) International Journal of Molecular Sciences, 21 (21), art. no. 7966, pp. 1-21.

- Aphid-Responsive Defense Networks in Hybrid Switchgrass

(2020) Frontiers in Plant Science, 11, art. no. 1145, .

Please also add novelty statement in the end of introduction.

A novelty statement has been inserted in the end of introduction.

Please add recent references in method section (Isothermal titration calorimetry and FRET).

Recent references for both approaches in the method section have been added.

Please improve discussion by addition of some recent relevant references supporting your findings.

After inserting six additional references following review 1’s suggestion, we feel most of the recent relevant references are cited in the revised manuscript.

Please also highlight your finding in discussion section and at the end of discussion add a short future perspective.

Discussion has been modified to accommodate reviewer’s concern.

Reviewer 3 Report

Comments and Suggestions for Authors

The article Structural and interactional analysis of the flavonoid biosynthetic proteins chalcone synthase, chalcone isomerase and chalcone isomerase like-proteions by Lewis et al. provides an exhaustive multimethodological study of the interaction of CHS, CHI and the CHI-like proteins, and includes crystal structures, kinetic studies and docking experiments. I strongly supports the metabolon Theory for flavonoid biosynthesis.

The article is well written and addresses an important topi. I have only a few minor comments:

line 2: Remove biosynthetic from the title

line 3: remove the colon in the title

line 14: remove synthetic

line 15: remove synthesis of flavonoid pigments. Of course the pathway synthesizes flavonoids, and those provide UV protection etc, flavonoid synthesis does not fit here, may be replace it by pigmentation

Fig.1: Fig 1 A is too small. Separate Fig. 1A and 1B intwo Fig. 1 and 2. Rearrange flavonols and flavones boxes. flavonols can go below, flavones may be up, this allows you to extend size

Fig.1B: turn chalcone position by 180° so that position of rings A and B reflect the positions in the flavanone structure

line 100: how can CHi provide 5-deoxyflavone?

line 103: 5-deoxyflavanone

line 186: malus domestica: italics

Author Response

Reviewer 3

The article Structural and interactional analysis of the flavonoid biosynthetic proteins chalcone synthase, chalcone isomerase and chalcone isomerase like-proteions by Lewis et al. provides an exhaustive multimethodological study of the interaction of CHS, CHI and the CHI-like proteins, and includes crystal structures, kinetic studies and docking experiments. I strongly supports the metabolon Theory for flavonoid biosynthesis.

The article is well written and addresses an important topic. I have only a few minor comments:

line 2: Remove biosynthetic from the title

biosynthetic has been changed to pathway

line 3: remove the colon in the title.

We think the colon can be removed without an impact on the title. However, the reason for leaving the colon in place is if there was/will be a series of publications, each on different enzymes. Thus, we wish to keep the current title.

line 14: remove synthetic

The word “Synthetic” has been removed.

line 15: remove synthesis of flavonoid pigments. Of course the pathway synthesizes flavonoids, and those provide UV protection etc, flavonoid synthesis does not fit here, may be replace it by pigmentation

The word “Pigmentation” has been inserted.

Fig.1: Fig 1 A is too small. Separate Fig. 1A and 1B in two Fig. 1 and 2. Rearrange flavonols and flavones boxes. flavonols can go below, flavones may be up, this allows you to extend size

Figure 1A has been updated following the suggestion.

Fig.1B: turn chalcone position by 180° so that position of rings A and B reflect the positions in the flavanone structure

Fig. 1B has been updated following the suggestion.

line 100: how can CHi provide 5-deoxyflavone?

line 103: 5-deoxyflavanone

In the revised manuscript, the corresponding terms in line 100 and 103 were corrected.

line 186: malus domestica: italics

It is italicized in the revised manuscript.

Reviewer 4 Report

Comments and Suggestions for Authors

The text below contains comments on manuscript entitled “Structural and interactional analysis of the flavonoid biosynthetic proteins: chalcone synthase, chalcone isomerase and chalcone isomerase like-protein”.

The manuscript is well written on sufficiently good English. The experimental design is well structured containing various methodologies used. The result and discussion section contains sufficient results to explain the experiments outcomes.

I have some minor suggestions for corrections listed:

Page 3, Line 92: Chalcone isomerase is not necessary to be in italic font and underlined.

To my opinion the introduction is too long. I am not sure is so many details about the enzymes are necessary. A clear and well defined aim of the study is missing.

Page 6, Line 186: The Latin name of the plants such as Malus domestica and Glycine max must be in italic font.

Comments on the Quality of English Language

 Minor editing of English language required

Author Response

Reviewer 4

The manuscript is well written on sufficiently good English. The experimental design is well structured containing various methodologies used. The result and discussion section contains sufficient results to explain the experiments outcomes.

I have some minor suggestions for corrections listed:

Page 3, Line 92: Chalcone isomerase is not necessary to be in italic font and underlined.

It has been corrected in the revised version.

To my opinion the introduction is too long. I am not sure is so many details about the enzymes are necessary. A clear and well defined aim of the study is missing.

As recommended by other reviewers, The introduction has been modified to make a significance and aim clearer.

Page 6, Line 186: The Latin name of the plants such as Malus domestica and Glycine max must be in italic font.

They have been updated in the revised version.

Round 2

Reviewer 2 Report

Comments and Suggestions for Authors

Manuscript revised according to suggestions